# Weak-to-strong Generalization via Formative Learning from Student Demonstrations & Teacher Evaluation

## Abstract

As Large Language Models (LLMs) exceed human capabilities, providing reliable human feedback for evaluating and aligning them, via standard frameworks such as Reinforcement Learning from Human Feedback, becomes challenging. This raises a fundamental question: *how can we leverage weaker (teacher) supervision to elicit the full capabilities of a stronger (student) model?* This emerging paradigm, known as Weak-to-Strong (W2S) generalization, however, also introduces a key challenge as the strong student may "overfit" to the weak teacher's mistakes, resulting in a notable performance degradation compared to learning with ground-truth data. We show that this overfitting problem occurs because learning with weak supervision implicitly regularizes the strong student's policy toward the weak reference policy. Building on this insight, we propose a novel learning approach, called Weak Teacher **Ev**aluation of Strong Student **De**monstrations or EvE, to instead regularize the strong student toward its reference policy. EvE's regularization intuitively elicits the strong student's knowledge through its own task demonstrations while relying on the weaker teacher to evaluate these demonstrations – an instance of formative learning. Extensive empirical evaluations demonstrate that EvE significantly outperforms existing W2S learning approaches and exhibits significantly better robustness under unreliable feedback compared to contrastive learning methods such as Direct Preference Optimization.

## 1 Introduction

Reinforcement Learning from Human Feedback (RLHF) Ouyang et al. (2022); Christiano et al. (2017) has been a canonical framework for steering language models (LMs) to align with human values based on human demonstrations. This framework has demonstrated impressive performance across a wide range of tasks, from conversation to coding, where humans "can" provide reliable supervision. In the future, as these AI models reach or exceed human capabilities, they will be capable of solving complex tasks that are difficult for humans to supervise. For example, when these AI models acquire the ability to generate a code project with millions of lines of code or summarize an entire book with thousands of pages, humans are unlikely to provide reliable feedback to align these superhuman AI models effectively.

*How can we align these superhuman AI models given the likely unreliable human supervision?* Burns et al. (2024) study this question by using a smaller LLM to represent unreliable human supervision on binary classification tasks. Effectively, this "weaker" teacher is prone to make mistakes when supervising a "stronger" student model. They observed a phenomenon called *weak-to-strong (W2S) generalization* – a stronger model finetuned with labels generated by a weaker model could outperform this weaker teacher without even seeing the ground truth labels. Despite the promising results, a key challenge in learning from weak supervision is the risk of overfitting Burns et al. (2024), where the strong student inevitably learns to imitate the errors of the weak teacher. Burns et al. (2024) study early-stopping as an implicit regularization to prevent overfitting, but notes that early-stopping does not constitute a valid method as it unrealistically requires ground-truth labels.

This paper first provides a crucial theoretical insight into the overfitting problem in W2S generalization. Specifically, by representing the weak teacher as an Energy-Based Model (EBM), we reveal that learning

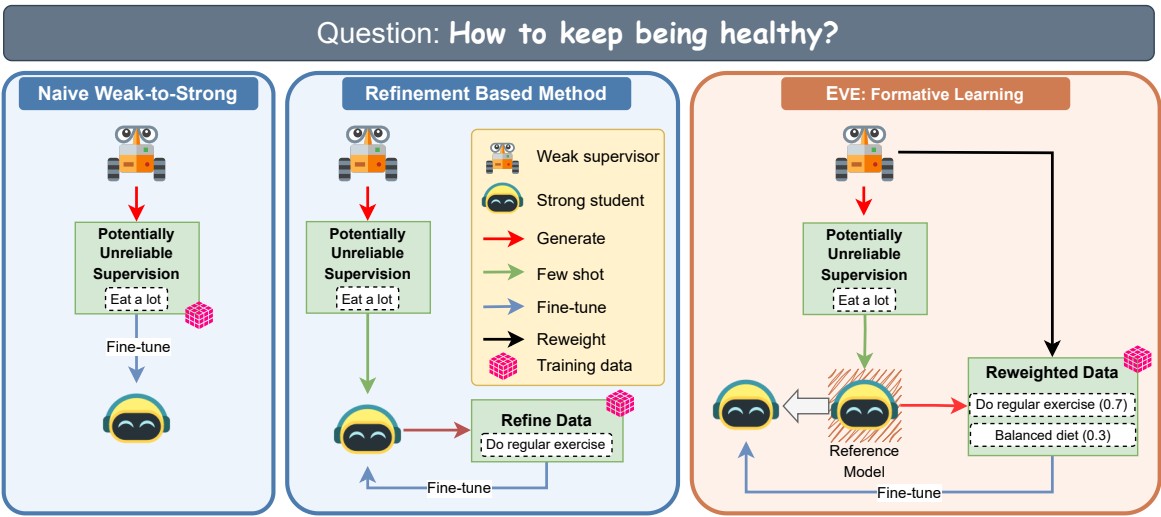

Figure 1: **Eve and existing W2S generalization methods**. Naive learning overfits the weak reference model, potentially imitating its mistakes (e.g., *"Eat a lot"*). Refinement learning "refines" the weak supervision (i.e., *"Do regular exercise"*). In contrast, Eve leverages the weak teacher as a reward function while eliciting the student's reference model salient knowledge

from weak supervision involves maximizing the reward while simultaneously regularizing the strong student's policy toward the weak reference model. This process leads to a drawback: the strong student not only inherits the informative supervision but also amplifies the errors of the weak teacher, ultimately degrading the student's overall performance on the desired tasks (Hong et al., 2024).

Building upon this insight, we propose a novel learning method, called Weak Teacher **Ev**aluation of Strong Student D**e**monstrations (Eve), to enable the strong student to elicit its own (prior) knowledge of the task while relying on the weak teacher to evaluate, or score, such demonstrations – an instance of formative learning, effectively utilizing both the knowledge of the weak teacher and the student's reference model. As depicted in Fig. 1, Eve utilizes the weak teacher's demonstrations to prompt the strong student, allowing it to generate its own training data reflecting its understanding of the tasks. The generated samples are then adjusted by the logarithmic ratio of the weak teacher's policy pre- and post-alignment, which serves as a reward signal to guide the strong student's learning.

In summary, (1) we provide a theoretical characterization of overfitting in W2S learning; then (2) we introduce Eve, an approach that enables learning from strong student demonstrations, where the weak teacher acts as a reward function to evaluate the strong student's outputs; finally, (3) we show that Eve significantly outperforms naive W2S learning by overcoming the overfitting issue, demonstrating the effectiveness of utilizing the strong student's critical thinking ability under the weak teacher's reward evaluation; surprisingly, when learning from a weak and unreliable reward signal, Eve – an off-policy method – achieves significantly better performance and robustness to contrastive learning methods such as DPO Rafailov et al. (2023).

## 2 Related Work

### 2.1 Weak-to-strong Generalization

Burns et al. (2024) introduce a synthetic setup to study whether a stronger model can generalize well with weaker supervision, compared to training with high-quality or ground-truth data. Prior efforts investigate W2S phenomena only in binary classification setups, leaving other practical alignment-relevant tasks (e.g., open-ended text generation whose output has no fixed length and requires sharing vocabulary size between the strong student and weak teacher) largely under-explored Ye et al. (2024); Cui et al. (2024); Agrawal et al.

(2024). Another line of work Somerstep et al. (2024); Ye et al. (2025); Zheng et al. (2024) leverages the pre-trained knowledge of the strong student to refine labels curated from the weak teacher, thereby improving the supervision quality. Ye et al. (2025) study W2S generalization on text-generation tasks, where they simulate *unreliable demonstrations* and *unreliable comparison feedback* during the alignment phase.

Different from the prior work, this paper extends W2S generalization beyond classification. We elicit the latent knowledge of the strong student about the intended tasks, which is then evaluated by the weak teacher's reward model. Additionally, by interpreting learning from weak supervision as reward maximization, our approach generalizes refinement-based methods Ye et al. (2025); Yang et al. (2024).
'

## 2.2 Reinforcement Learning from Human Feedback

RLHF aims to align LMs with human preferences and values (Christiano et al., 2017; Bai et al., 2022), and has demonstrated impressive performance on established benchmarks (OpenAI et al., 2024; Hugo Touvron, 2023; Xiong et al., 2024a;b; Wang et al., 2024). However, the RLHF pipeline incurs significant computational costs and requires a large amount of high-quality human preference labels.

Recent advancements, such as Direct Alignment Algorithms (DAAs) (Rafailov et al., 2023; Tang et al., 2024), bypass the need for an explicit reward model and directly train the LMs on the human preference data. Reinforcement Learning with AI Feedback Pang et al. (2024) uses a well-trained language model (e.g., GPT-4 or Claude-3.5 Sonnet) to provide preference feedback as a substitute for human supervision. More recently, Ye et al. (2025) study whether standard RLHF remains effective under unreliable feedback.

We demonstrate that **contrastive-learning** approaches (Rafailov et al., 2024; Azar et al., 2024) heavily suffer from the reward over-optimization issue Rafailov et al. (2024); Gao et al. (2023). In contrast, EVE – also an offline supervised approach – is significantly more robust to unreliable feedback and achieves a better reward-KL tradeoff than DAAs. This finding is significant as it contradicts observations in prior work (Tajwar et al., 2024), which shows that DAAs with negative gradient perform significantly better than offline supervised methods in conventional alignment scenarios with human feedback.

## 2.3 Reward Maximization with KL Regularization as Distributional Matching

Prior works show that reward maximization with KL regularization in standard RLHF can be viewed as minimizing the reverse KL between the LM policy $\pi_\theta$ and the target distribution that represents the aligned language model (Korbak et al., 2022b;a; Go et al., 2023). Other studies also explored the use of forward KL, which corresponds to setting the reward maximization as supervised learning (Norouzi et al., 2016; Peters and Schaal, 2007). Similarly, our paper shows that imitating a weak teacher can be viewed as reward maximization, where the reward is defined as the log probability of the weak teacher, with a KL regularization toward the weak reference model, causing the over-optimization problem.

# 3 Preliminaries

## 3.1 LLM Alignment with Human Preferences

LLM alignment can be viewed as reward-maximization with KL-constrained:

$$\max_{\pi_\theta} \mathbb{E}_{x \sim \mathcal{D}, y \sim \pi_\theta(\cdot|x)} \left[ r(x, y) \right] - \beta \mathrm{KL}(\pi_\theta || \pi_{\mathrm{ref}}) \tag{1}$$

where $y$ is a sampled response from $\pi_\theta$, $\beta$ controls the trade-off between maximizing the reward and deviation from the reference model $\pi_{\mathrm{ref}}$, and $r$ is the reward function that captures human preferences.

## 3.2 Offline Fine-Tuning Methods for Reward Maximization

Directly optimizing the objective in Eq. equation 1 requires repeated sampling, which is computationally expensive. Alternatively, equivalent offline methods fall into 2 main categories:

**Contrastive Learning Methods.** Approaches, such as DPO Rafailov et al. (2023) and IPO Azar et al. (2024), directly update the LM policy $\pi_\theta$ on human preference data. These methods represent the reward implicitly via the LM $\pi_\theta$ and the reference model $\pi_{\text{ref}}$ as:

$$r_\theta(x, y) = \beta \log \frac{\pi_\theta(y|x)}{\pi_{\text{ref}}(y|x)} + \beta \log Z(x) \tag{2}$$

where $Z(x) = \sum_y \pi_{\text{ref}}(y|x) \exp\left(\frac{1}{\beta} r_\theta(x, y)\right)$ is the normalization factor. Using this representation, a general objective can be derived to train the policy on human preference data, as follows:

$$\mathcal{L}(\pi_\theta, \pi_{\text{ref}}) = -\mathbb{E}_{(x, y_w, y_l) \sim \mathcal{D}} \left[ f\left( \beta \log \frac{\pi_\theta(y_w|x)}{\pi_{\text{ref}}(y_w|x)} - \beta \log \frac{\pi_\theta(y_l|x)}{\pi_{\text{ref}}(y_l|x)} \right) \right] \tag{3}$$

where $f$ is a convex loss function. The gradient of contrastive learning approaches, therefore, consists of a **positive gradient** term that increases the likelihood of the preferred response $y_w$ and a **negative gradient** term that pushes down the likelihood of the non-preferred response $y_l$.

**Offline Supervised Methods.** This alternative class of methods, including RAFT (Dong et al., 2023) and RWR (Peters and Schaal, 2007), minimizes a weighted maximum likelihood objective. Formally, these methods first sample $K$ completions per prompt $x$ from the reference model $\pi_{\text{ref}}$, i.e., $y_{1,\cdots,K} \sim \pi_{\text{ref}}(\cdot|x^{(i)})$. These responses are then weighted by a non-negative weighting function $F\left(x, y_k|y_{1,\cdots,K}\right)$ conditioned on the other sampled responses and maximize:

$$\max_{\pi_\theta} \mathbb{E}_{(x, y_{1,\cdots K}) \sim \mathcal{D}, \pi_{\text{ref}}(\cdot|x)} \left[ \sum_{k=1}^K \log \pi_\theta(y_k|x) \cdot F\left(x, y_k|y_{1,\cdots,K}\right) \right]$$

Intuitively, since $F\left(x, y|y_{1,\cdots K}\right)$ is always non-negative, these methods always increase the likelihood of responses generated from $\pi_{\text{ref}}$. Responses that are more preferred will be assigned higher weights, there is no **negative gradient** effect to push down the likelihood of suboptimal responses.

### 3.3 Weak-to-Strong Evaluation Pipeline

We review the W2S evaluation pipeline in (Burns et al., 2024), which consists of three stages, as follows:

**(1) Weak Teacher Creation:** The weak teacher is created by fine-tuning a small pre-trained model to align with human preferences. We utilize SFT+DPO, a standard preference learning pipeline, to ensure the weak model acquires knowledge about alignment tasks. The resulting model is denoted as $\pi^{\text{weak}}$.

**(2) Strong Student Learning with Weak Supervision:** The weak model is then used to generate weak supervision data $\mathcal{D}_{\text{weak}} = \{x^{(i)}, y^{(i)}\}$ where $x^{(i)}$ and $y^{(i)}$ are the prompt and the generated response from $\pi^{\text{weak}}$, respectively. The strong model $\pi_\theta$ is then fine-tuned using the weak supervision data with the SFT objective.

**(3) Strong Student Learning with Ground-truth Supervision:** Another strong model $\pi^{\text{strong}}$ is fine-tuned with the Ground-truth human labels to establish the upper-bound performance. To ensure that this aligned model fully acquires the target task's capabilities, it goes through an additional, preference learning phase (e.g., DPO).

The W2S generalization performance of $\pi_\theta$ can be measured by Performance Gap Recovered (**PGR**):

$$\text{PGR} = \frac{\mathcal{P}_{\text{weak-to-strong}} - \mathcal{P}_{\text{weak}}}{\mathcal{P}_{\text{strong}} - \mathcal{P}_{\text{weak}}}$$

where $\mathcal{P}_{\text{weak-to-strong}}$, $\mathcal{P}_{\text{weak}}$, and $\mathcal{P}_{\text{strong}}$ are the task performance of $\pi_\theta$, $\pi^{\text{weak}}$, and $\pi^{\text{strong}}$, respectively.

## 4 Formative Learning with Eve

### 4.1 Learning from Weak Supervision Implicitly Aligns with Weak Reference Model

This section connects W2S learning to reward maximization and builds the theory behind the model's behavior, i.e., its generalization characteristics.

We begin by representing the weak teacher in the form of energy-based models Rafailov et al. (2023); Levine (2018); Haarnoja et al. (2017):

$$\pi^{\text{weak}}(y|x) = \frac{1}{Z(x)} \pi_{\text{ref}}^{\text{weak}}(y|x) \exp\left(r^{\text{weak}}(x, y)/\beta\right)$$

where $\pi_{\text{ref}}^{\text{weak}}$ is the SFT version of $\pi^{\text{weak}}$.

**Proposition 4.1.** *W2s generalization with a weak teacher $\pi^{weak}(y|x)$ and a strong student $\pi_\theta$ (the training model) can be cast as the following optimization problem:*

$$\min_{\pi_\theta} KL\left(\pi^{weak}||\pi_\theta\right) \tag{4}$$

$$s.t \ \pi^{weak} = \arg\min_{\pi} KL\left(\pi||\pi^{EBM}\right)$$

*where $\pi^{EBM}(y|x) \propto \pi_{ref}^{weak}(y|x) \exp\left(r(x, y)/\beta\right)$.*

The proof is straightforward and deferred to the Appendix 11.2. This shows that imitating the weak teacher can be seen as finding an EBM policy $\pi^{\text{EBM}}$, which is the optimal solution in the lower-level objective. This leads to the following theorem.

**Theorem 4.2.** *The optimal solution to W2S generalization is equivalent to the optimal solution in the following objective:*

$$\max_{\pi_\theta} \mathbb{E}_{x \sim \mathcal{D}, y \sim \pi_\theta(\cdot|x)} \left[r^{weak}(x, y)\right] - \lambda KL(\pi_\theta||\pi_{ref}^{weak}) \tag{5}$$

***Proof Sketch.*** Notice that the objective for training the strong student, and the reverse KL share the same optimal solution $\pi_\theta$. In addition, it can be shown that minimizing the reverse KL between the strong student and the weak teacher,

$$\min_{\pi_\theta} \text{KL}\left(\pi_\theta||\pi^{\text{weak}}\right), \tag{6}$$

is equivalent to maximizing the KL-constrained reward objective in Eq. equation 5. □

Theorem 4.2 provides a key insight: imitating the weak teacher maximize an implicit reward, $r^{\text{weak}}(x, y) = \beta \log \pi^{\text{weak}}(y|x) - \beta \log \pi_{\text{ref}}^{\text{weak}}(y|x)$, while regularizing (with KL objective) the strong student toward the weak reference model $\pi_{\text{ref}}^{\text{weak}}$. Consequently, instead of aiming to elicit knowledge of the strong student, existing W2S learning remains confined to the knowledge of the weak model, which may adversely impact the strong student's performance.

### 4.2 Suboptimal Weak-to-Strong Generalization toward Weak Reference Model

We empirically confirm the theoretical insight in the previous section. Specifically, we analyze the W2S training progression on $\mathcal{D}_{\text{weak}}$: at each checkpoint, we generate responses using the corresponding intermediate model with the same set of prompts, from which we calculate the implicit reward $r^{\text{weak}}(x, y) = \beta \log \pi^{\text{weak}}(y|x) - \beta \log \pi_{\text{ref}}^{\text{weak}}(y|x)$, the divergence $\text{KL}(\pi_\theta||\pi_{\text{ref}}^{\text{weak}})$, and the PGR.

Fig. 2 shows that while the strong model learns to maximize the implicit reward (Left), the learned policy is also regularized towards the weak reference model $\pi_{\text{ref}}^{\text{weak}}$, indicated by the consistently low KL divergence $\text{KL}(\pi_\theta||\pi_{\text{ref}}^{\text{weak}})$ shortly after the training progresses (Right). Moreover, we also observe that the PGR, as measured by the golden reward function, decreases significantly (Right). This suggests that imitating the weak reference model $\pi_{\text{ref}}^{\text{weak}}$ (and potentially inheriting its mistakes) negatively impacts the performance of the strong student.

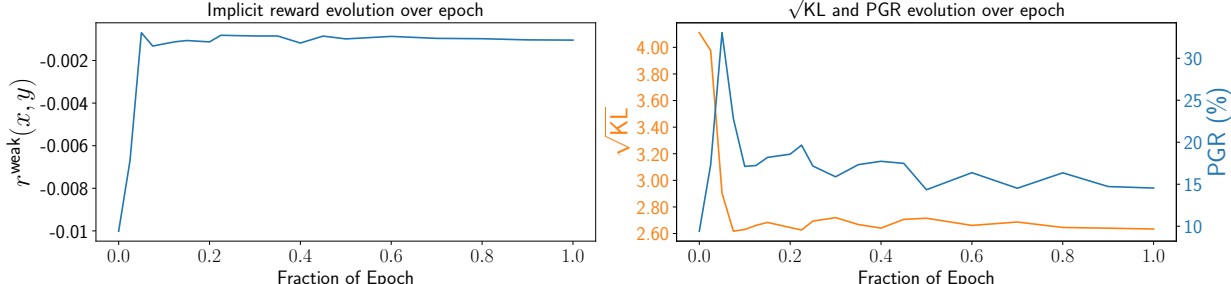

Figure 2: **Learning from weak supervision** as reward maximization. **Left:** the strong model $\pi_\theta$ learns to maximize the implicit reward $r^{\text{weak}}(x, y) = \beta \log \pi_{\text{align}}^{\text{weak}}(y|x) - \beta \log \pi_{\text{ref}}^{\text{weak}}(y|x)$. **Right:** the strong model also learns to imitate the weak reference model $\pi_{\text{ref}}^{\text{weak}}$'s mistakes, leading to performance degradation (in PGR).

### 4.3 Eve: Eliciting Strong Student Knowledge

Motivated by the connection between imitating the weak teacher and reward maximization, we "generalize" the KL-constrained reward maximization learning of the strong student $\pi$:

$$\max_{\pi_\theta} \mathbb{E}_{x \sim \mathcal{D}, y \sim \pi_\theta(\cdot|x)} \left[ r^{\text{weak}}(x, y) \right] - \lambda \text{KL}(\pi_\theta || \hat{\pi}) \tag{7}$$

where $\lambda$ controls the trade-off between maximizing the reward and deviation from a regularization policy $\hat{\pi}(y|x)$. Next, we propose one specific choice of the regularization policy $\hat{\pi}$ that can facilitate the elicitation of the strong student's knowledge, thereby enhancing W2S generalization.

**The choice of regularization policy $\hat{\pi}$.** Burns et al. (2024) interpret W2S generalization in terms of saliency: some tasks are already salient to the strong student; in this view, the role of the weak teacher is to elicit the student's latent knowledge rather than enforcing naive imitation of the weak teacher's own demonstrations. Inspired by this interpretation, we propose to regularize the learning policy toward the strong student pre-trained model, i.e., $\hat{\pi}(y|x) = \pi_{\text{ref}}^{\text{strong}}(y|x)$. This design choice serves an important goal: to encourage the learned policy $\pi_\theta$ to remain close to the initial strong reference model $\pi_{\text{ref}}^{\text{strong}}$, thereby facilitating the elicitation of the student's prior knowledge while simultaneously incorporating assessment from the weak teacher. Similar to Burns et al. (2024), to elicit the strong student's knowledge of the task, we first create the weak teacher's demonstrations, which are then used in few-shot prompting the strong reference model $\pi_{\text{ref}}^{\text{strong}}$ to generate task-relevant outputs, as $\pi_{\text{ref}}^{\text{strong}}$ is not trained to follow instructions. We provide detailed examples in Appendix 9.6.

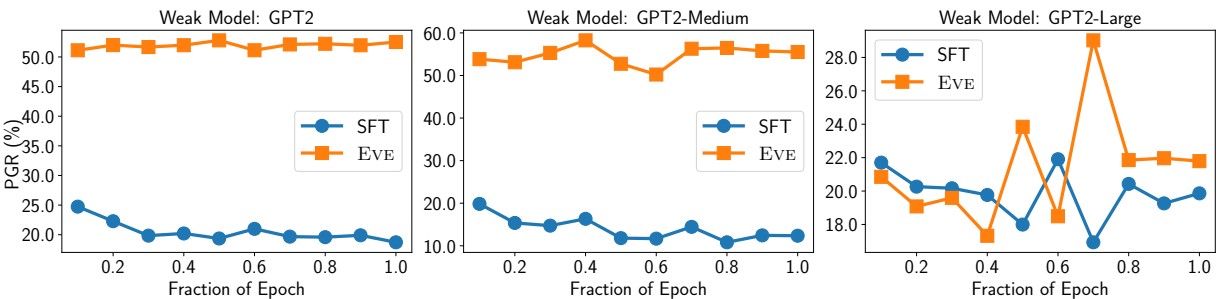

Figure 3: Evolution of PGR (%). We observe clear signs of overfitting to the weak teacher's errors well before finishing a single epoch. Notably, when there is a large gap between the strong student and the weak teacher, the student reaches its best performance within the first 10% of the epoch. EVE has little to no PGR degradation and significantly outperforms naive W2S learning (SFT).

**Optimization.** Directly optimizing the objective in Eq. equation 7 can incur significant computational costs, as it requires repeated sampling from the strong student $\pi_\theta$ inside the training loop Rafailov et al. (2023). Following prior work (Rafailov et al., 2023; Peters and Schaal, 2007; Peng et al., 2019), it is straightforward to show that the optimal policy to this KL-constrained objective takes the form:

$$\pi_r(y|x) = \frac{1}{Z(x)} \exp\left(r^{\text{weak}}(x, y)/\lambda\right) \pi_{\text{ref}}^{\text{strong}}(y|x)$$

where $Z(x) = \sum_y \pi_{\text{ref}}^{\text{strong}}(y|x) \exp\left(r(x, y)/\lambda\right)$ is the normalization constant. We can also leverage the duality between the reward function and the weak teacher $\pi^{\text{weak}}$ (Rafailov et al., 2023). Given the optimal policy $\pi_r$, we can then formulate a supervised learning objective for the parametrized strong student $\pi_\theta$ to match with this optimal policy, resulting in the following objective:

$$\max_{\pi_\theta} \mathcal{J}(\pi_\theta) = \max_{\pi_\theta} \mathbb{E}_{x\sim\mathcal{D}, y\sim\pi_{\text{ref}}^{\text{strong}}(\cdot|x)} \left[ \frac{\left(\pi^{\text{weak}}(y|x)/\pi_{\text{ref}}^{\text{weak}}(y|x)\right)^{\beta/\lambda}}{Z(x)} \cdot \log \pi_\theta(y|x) \right]$$

where the $\beta/\lambda$ ratio controls the impact of the weak-supervision reward signal during the strong student's updates. A high $\beta/\lambda$ ratio leads to a more uniform update, where all samples are assigned similar weights; i.e., there will be no weak supervision in learning. Conversely, a low $\beta/\lambda$ ratio results in a more focused policy update that prioritizes samples with high weak-supervision reward signals. This objective avoids sampling directly from $\pi_\theta$ on every update as $\pi_\theta$ changes during training; instead, we can sample the responses from the fixed $\pi_{\text{ref}}^{\text{strong}}$ once at the beginning of the optimization, which is significantly more efficient.

We also estimate the intractable normalization factor $Z(x)$ using *Self-Normalizing Importance Sampling* (Owen, 2013). Formally, given $K > 1$ i.i.d. completions $y^1, \cdots, y^N \sim \pi_{\text{ref}}^{\text{strong}}(\cdot|x)$ drawn from strong reference model, we can define an empirical distribution by normalizing the log-ratio $f(x, y) = \frac{\beta}{\lambda}\left(\log \pi^{\text{weak}}(y|x) - \log \pi_{\text{ref}}^{\text{weak}}(y|x)\right)$ over $K$ samples:

$$F(x, y^i|y^{1,\cdots,K}) = \frac{K \cdot \exp\left(f(x, y^i)\right)}{\sum_{k=1}^{K} \exp\left(f(x, y^k)\right)} \tag{8}$$

where the normalization is estimated by $Z(x) \approx \frac{1}{K}\sum_{k=1}^{K} \exp\left(f(x, y^k)\right)$. In summary, the final estimate is:

$$\mathcal{J}(\pi_\theta) = \mathbb{E}_{x\sim\mathcal{D}, y^{1,\cdots,K}\sim\pi_{\text{ref}}^{\text{strong}}(\cdot|x)} \left[\log \pi_\theta(y^i|x) \cdot F(x, y^i|y^{1,\cdots,K})\right]$$

We refer to this W2S learning approach as EVE. EVE can be seen as an offline supervised method, where the weighting function is the exponential of the implicit reward defined in Eq. equation 2.

## 5 Experiments

In this section, we empirically evaluate EVE's W2S generalization performance on two tasks: **controlled-summarization** and **instruction following**.

### 5.1 Controlled-Summarization

**Setup.** We choose the representative Reddit TL;DR summarization Stiennon et al. (2020) dataset and follow the synthetic setup from Gao et al. (2023); Zhou et al. (2024); Rafailov et al. (2023), where we train a *golden* reward model $r_{\text{gold}}(x, y)$ to label synthetic preference data $\mathcal{D}_{\text{golden}}$ for fine-tune weak-aligned model and evaluation. We use GPT2-series (Radford et al., 2019) (GPT2-Base/Medium/Large) as weak teachers and a more advanced LLama-3.2-3B model (MetaAI, 2024a;b) as the strong student. The weak model $\pi^{\text{weak}}$ is the aligned model with DPO (Rafailov et al., 2023) from $\mathcal{D}_{\text{golden}}$.

**Baselines.** In additional to EVE, we evaluate several existing W2S approaches, including **SFT** – which naively fine-tunes the strong student on weak supervision data $\mathcal{D}_{\text{weak}}$ – and (2) **Refinement** (Somerstep

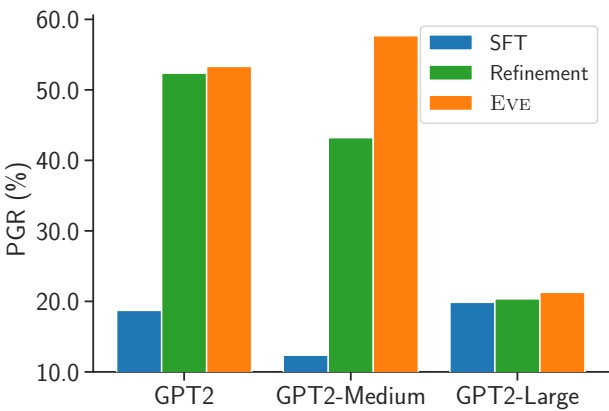

Figure 4: PGR (%) of SFT, Refinement and EVE.

et al., 2024; Yang et al., 2024) – which prompts the strong student to refine the responses generated by the weak teacher and fine-tunes the strong student with the refined responses.

**Results.** Fig. 4 shows the PGR results. EVE consistently outperforms the other baselines across all weak teachers. Notably, under the supervision of GPT-2 (the weakest model), EVE achieves a nearly 25% performance boost over SFT. Moreover, SFT achieves the peak performance early in training (around 10% of the epoch), but its performance steadily declines thereafter. In contrast, **Eve demonstrates minimal to no degradation in PGR over the course of the training process**. As discussed in Section 4, this can be attributed to the ability of EVE to more effectively balance learning from the weak teacher and the salient knowledge of the strong reference model.

**Impact of $\beta/\lambda$ ratio.** We investigate the impact of $\beta/\lambda$ on W2S performance. Fig. 5 illustrates the impact of $\beta/\lambda$ on PGR across different weak teachers. Setting $\beta/\lambda$ around 1.0 achieves optimal or near-optimal performance. Consequently, we default $\beta/\lambda = 1.0$ in all experiments, **eliminating the need for hyperparameter tuning that requires ground-truth labels**. Without the weak supervision (i.e., $\beta/\lambda = \infty$), the performance significantly decreases; this confirms the benefit of learning from the weak teacher's reward signals. Conversely, setting $\beta/\lambda$ to a very low value can also degrade the performance. One possible explanation is that, as $\beta/\lambda \to 0$, the weighting function $F(x, y^i | y^{1,\cdots,K})$ converges to a one-hot distribution, where the response with the highest reward is assigned a weight of 1 and the rest are ignored. This limits learning from a few samples, making it susceptible to simply memorizing the training data (Park et al., 2024).

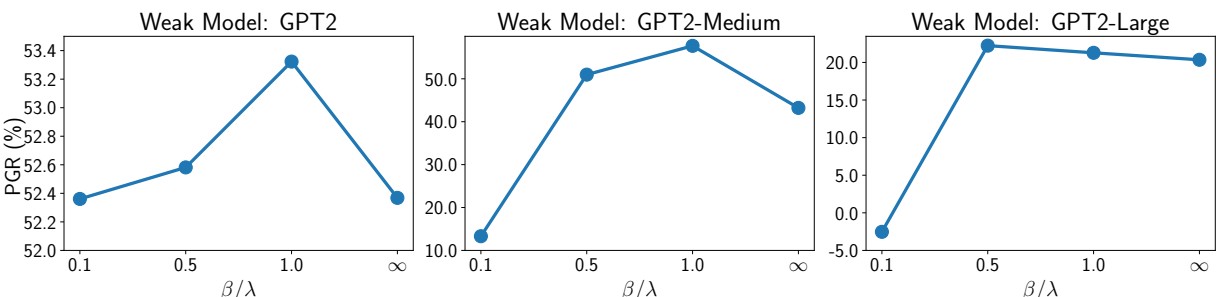

Figure 5: PGR (%) of various $\beta/\lambda$ ratios in EVE's objective.

**Scaling dataset size.** We additionally study the impact of scaling the number of responses $K$ per prompt. Fig. 6 shows the performance of EVE and SFT. EVE demonstrates improved performance as we increase the size of the training dataset (especially as the weak teacher is stronger), while SFT's performance decreases. This can be explained by the fact that as the training data size increases, the strong student also becomes

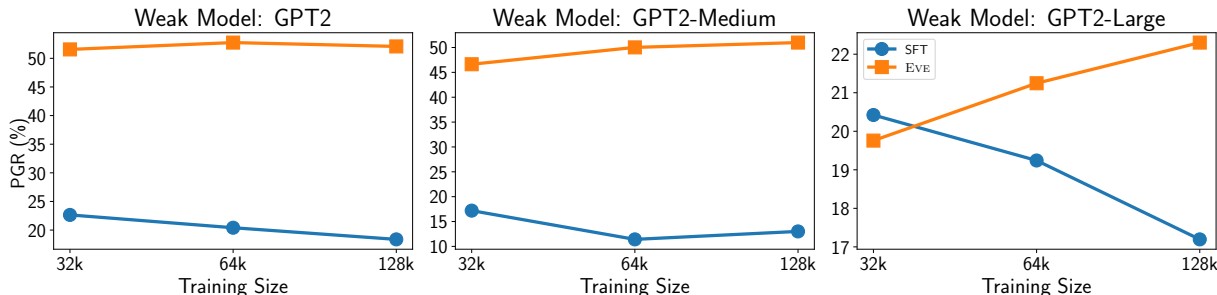

Figure 6: Scaling the training size (32k, 64k and 128k) in EVE and SFT (trained for one epoch). EVE shows notable improvement as the training size increases, while SFT suffers from overfitting.

more susceptible to learning the weak teacher's mistakes. In contrast, EVE is designed to avoid this overfitting problem, thus, it can leverage the increased supervision significantly better.

## 5.2 Instruction Following

**Setup.** We use Qwen2.5-7B as the strong student and Llama-3.2-1B as the weak teacher. The strong reference model $\pi_{\text{base}}^{\text{strong}}$ is initialized from the pre-trained distribution, and the weak model $\pi^{\text{weak}}$ is fine-tuned with DPO on the UltraFeedback dataset (Ding et al., 2023).

**Evaluation.** We evaluate EVE on two standard instruction-following benchmarks, AlpacaEval 2.0 (Dubois et al., 2025) and IFEval (Zhou et al., 2023). For AlpacaEval 2.0, we report length-controlled win-rates against `gpt4-turbo`, with `gpt-4o-mini` serving as the judge.

**Baselines.** We evaluate EVE against **SFT, Refinement** and **DPO** - which uses the weak teacher as reward signal to label preference data generated by the strong student. Following prior works (Rafailov et al., 2024; Gao et al., 2024), we train DPO for 1 epoch with $\beta = 0.05$ as default hyperparameters.

**Results.** We report the results in Fig. 7. EVE consistently outperforms the other W2S approaches across all benchmarks. Interestingly, we find that weak supervision can provide a reliable signal for guiding the strong student, not only in encouraging instruction-following behavior but also helping to filter out non-compliant responses.

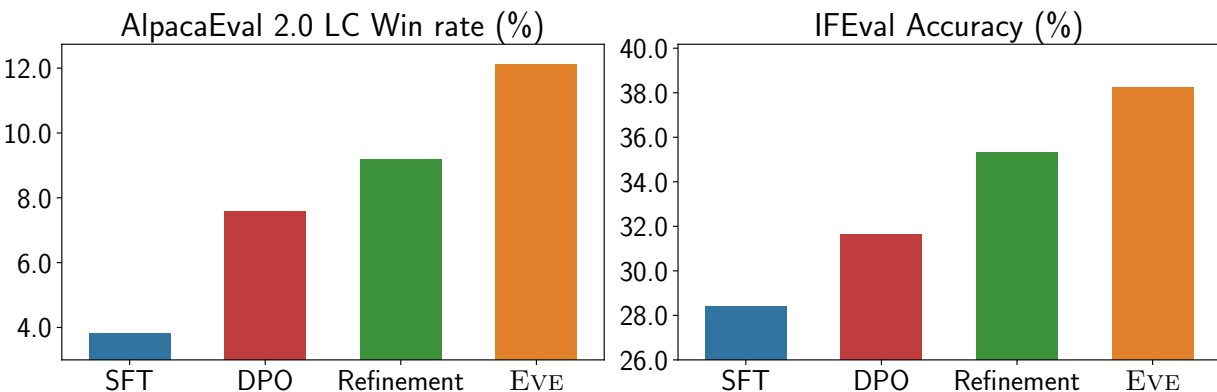

Figure 7: Win rate and Accuracy on 2 standard instruction-following benchmarks, AlpacaEval 2.0 and IFEval, respectively.

### 5.3 MLE and Contrastive Learning in W2S Generalization

Fig. 7 also shows the advantages of EVE over contrastive learning approaches (e.g., DPO). Standard RLHF frameworks (e.g., PPO (Schulman et al., 2017) and DPO (Rafailov et al., 2023)) can be seen as optimizing the reverse KL, while EVE optimizes the forward KL. As noted in (Tajwar et al., 2024), the reverse KL can modify the probability mass more aggressively than forward KL, resulting in a large deviation from $\pi_{\text{ref}}$ to find the peak reward region. Conversely, the forward KL tends to deviate less from its initial distribution towards the peak reward. This might be beneficial in W2S learning due to the following reasons:

**(i) Unreliable learning signal**. Unlike standard RLHF, W2S's feedback is highly unreliable. Finding the peak reward region with the reverse KL can result in performance degradation due to over-optimization as observed in Ye et al. (2025) (and re-confirmed in Fig 9 in our Appendix). Importantly, over-optimization can be more severe in W2S generalization as even humans cannot providea reliable signal to avoid these undesirable behaviors (Burns et al., 2024).

**(ii) Already capable strong student.** Similar to prior works (Burns et al., 2024; Ye et al., 2025; Cui et al., 2024), we assume that the strong student is already capable of solving the target tasks. Consequently, we hypothesize that the response region that achieved high rewards should be near the strong student. Therefore, the forward KL, inducing less deviation from the initial distribution, can be seen as an additional implicit regularization and performs better.

## 6 Limitations

We did not experiment with larger language models ($> 7$B) due to limited computational resources. Given the resource demands of generating data from the strong student, future work will focus on using the strong student for evaluation to eliminate the need for generating strong student data. For example, one direction is to explore extensions to our strategy proposed in Section 11.1, which relies on the strong student only for reward evaluation. Tajwar et al. (2024). While our method does introduce additional memory overhead from teacher feedback calculations, this cost is relatively minimal compared to the overall training process of the strong student.

## 7 Conclusion and Discussion

This paper studies the W2S generalization and provides a new theoretical perspective on imitating the weak teacher. We show that imitating the weak teacher is equivalent to maximizing an implicit reward and regularizing the student towards the weak reference policy, which can amplify the bias or mistakes of this supervised fine-tuned weak teacher while not effectively eliciting knowledge from the strong student. Building upon this observation, we propose EVE, which directly optimizes the strong student using an RLHF objective with the "forward KL" regularization towards its latent knowledge of the given task. Extensive empirical results demonstrate that EVE achieves superior performance to existing W2S baselines and effectively mitigates the overfitting problem in W2S generalization.

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

# 8 Societal Impacts

Our work demonstrates a positive societal impact with better alignment with human values, including helpfulness and harmlessness. We do not expect any negative societal impacts directly resulting from the contributions presented in our paper.

## 8.1 Computing Resources Specification

We train and evaluate our models using 4xNVIDIA H100 GPUs. All evaluations are judged by "gpt-4o-mini", with random positional flips to avoid position bias.

# 9 Further Details on the Experimental Setup

## 9.1 Gold-reward training details

We follow the synthetic setup in which we use the gold reward models to play the roles of humans and provide preference feedback (Gao et al., 2023; Zhou et al., 2024).

The golden reward model is initialized from Llama-3.1-8B. We first pool together the human preference Reddit TL;DR summarization dataset. We, then, apply SFT on the SFT split in Reddit TL;DR dataset. We then fine-tune the golden reward on the preference dataset. For both SFT and reward modeling phases, we use a batch size of 128, a learning rate of $1e-6$ over one epoch with a cosine learning rate schedule with 150 warm-up steps. The golden reward model, then, generates synthetic preferences on the original dataset with $p(y_1 \succ y_2 \mid x) = \sigma(r_{\text{gold}}(x, y_1) - r_{\text{gold}}(x, y_2))$, where $\sigma$ is the sigmoid function. The gold reward models achieve high test accuracies on summarization tasks with $75.8\%$ accuracy, showing a strong correlation with human preferences.

## 9.2 Direct Alignment Algorithms (DAAs) for weak models training setups

We apply DAAs on the synthetic golden preferences $\mathcal{D}_{\text{golden}} = \{x^{(i)}, y_w^{(i)}, y_l^{(i)}\}_{i=1}^N$, which involves two stages: Supervised Fine-Tuning (SFT) and Direct Preference Optimization (DPO) Rafailov et al. (2023). During SFT, the weak models are trained on *chosen responses only*. During DPO, we use standard setting of $\beta = 0.1$. For both phases, we use a batch size of 128, a learning rate of $1e-6$, and a cosine learning rate scheduler with a warmup of 150 steps.

## 9.3 Details of estimating KL divergence

We can construct an unbiased estimate of $\text{KL}(\pi_\theta || \pi_{\text{ref}})$ by sampling. Specifically, we first sample $N$ prompts $\{x^{(i)}\}_{i=1}^N$ from the evaluation set, and then for each prompt $x^{(i)}$, we sample a response $y^{(i)} \sim \pi_\theta(\cdot | x^{(i)})$ from the learned policy $\pi_\theta$. We can estimate the KL divergence at the sequence level as follows:

$$\frac{1}{N} \sum_{i=1}^N \log \pi_\theta(y^{(i)} \mid x^{(i)}) - \log \pi_{\text{ref}}(y^{(i)} \mid x^{(i)})$$

However, this estimation has high variance and can be negative. Therefore, we use the following unbiased estimator Schulman (2020):

$$\frac{1}{N} \sum_{i=1}^N \left( \frac{\pi_{\text{ref}}(y^{(i)} \mid x^{(i)})}{\pi_\theta(y^{(i)} \mid x^{(i)})} - 1 \right) + \log \frac{\pi_\theta(y^{(i)} \mid x^{(i)})}{\pi_{\text{ref}}(y^{(i)} \mid x^{(i)})}$$

## 9.4 Simulating Weak Supervision on Reward Modeling

**Models**: The reward model is first initialized from the supervised fine-tuned stage in RLHF, we remove the unembedding layer of the model and add a linear head to output a scalar value, which represents the reward for model completion.

**Hyperparameters**: We use a learning rate of $1e - 6$, and a cosine learning rate scheduler for training the reward model with a batch size of 64 and a warmup of over 150 steps.

**Weak labels**: We train a proxy reward model on half of golden preference dataset $\mathcal{D}_{\text{golden}}$ that is labeled by the golden reward model and then generate weak labels on the other half to create a weak preference data $\mathcal{D}_{\text{pref}}$ The weak label preference dataset is used to train a new reward model $\tilde{r}(x, y)$ for RL fine-tuning stage.

### 9.5 Detailed Calculation of Performance Gap Recovered (PGR)

To evaluate the performance $\mathcal{P}$ of the LM policy $\pi$, we use the golden reward model to score $\pi$ generated responses. Specifically, given a policy , its performance is defined as follows:

$$\mathcal{P}(\pi) = \mathbb{E}_{x \sim \mathcal{D}_{\text{test}}, y \sim \pi(\cdot|x)} \left[ r_{\text{gold}}(x, y) \right] \approx \frac{1}{N} \sum_{i=1}^{N} r_{\text{gold}}(x, y)$$

where $\mathcal{D}_{\text{test}}$ is an evaluation set of prompts. Performance Gap Recovered (PGR) of a W2S policy $\pi_\theta$ is calculated by:

$$\text{PGR} = \frac{\mathcal{P}(\pi_\theta) - \mathcal{P}(\pi^{\text{weak}})}{\mathcal{P}(\pi^{\text{strong}}) - \mathcal{P}(\pi^{\text{weak}})}$$

For all experiments, we set $N = 256$, similar to prior works (Rafailov et al., 2023; 2024).

### 9.6 Prompt Template for Refinement Sampling from Strong Models

**Controlled-summarization:** We use 2-shot prompting to elicit strong student knowledge. To be more specific, we randomly sample 2 prompts on a held-out set of prompts dedicated only to training the strong student. These 2 prompts are input to the weak aligned model to generate 2 responses. We then use the two responses as a demonstration using the following format:

examplar[1].promptTL;DR:    examplar[1].response    examplar[2].promptTL;DR:    examplar[2].response promptTL;DR:    **Instruction following.**    For the instruction following task, we ask the strong student to refine the weak teacher's response to align with the helpfulness objective with the following format:

User: examplar[1].prompt Weak Assistant: examplar[1].response Can you make the response more helpful and coherent? Strong Assistant:

## 10   A Bayesian Interpretation of W2S Generalization

Burns et al.Burns et al. (2024) saliency interpretation can essentially be viewed as a Bayesian inference problem. Intuitively, Bayesian inference is the problem of updating a distribution to conform to new evidence.

In W2S generalization setup, we update the strong student $\pi_\theta$, which is initialized from a prior $\pi_{\text{ref}}^{\text{strong}}$, that represents its knowledge about the tasks to conform with the evidence provided by the weak teacher data. The weak teacher data provides evidence about the intended tasks that we want the strong model to solve but also contains systematic errors of the weak teacher. Then, we can define the posterior distribution:

$$\pi^*(y|x) = \frac{1}{Z(x)} \pi^{\text{weak}}(y|x) \pi_{\text{ref}}^{\text{strong}}(y|x)$$
$$= \frac{1}{Z(x)} \exp\left(r'(x, y)/\beta\right) \pi_{\text{ref}}^{\text{strong}}(y|x)$$

Interestingly, this coincides with the optimal solution of the KL-constrained reward maximization objective where the reward is defined by $r'(x, y) = \log \pi^{\text{weak}}(y|x)$. Intuitively, rather than directly training the strong student to imitate the weak teacher can also lead to imitating the errors of the teacher. The posterior formulation avoids this failure mode by reinforcing the strong student to remain close to its initial knowledge while still learning the tasks effectively.

## 11 Mathematical Derivations

**Lemma 11.1.** *Consider an optimal policy such that there exists a minimizer to the following optimization problem:*

$$\min_{\pi_\theta} KL\left(\pi^{weak}||\pi_\theta\right)$$

$$s.t \; \pi^{weak}(y|x) = \arg\min_{\pi} KL(\pi||\pi^{EBM})$$

*where $\pi^{EBM}(y|x) \propto \pi_{ref}^{weak}(y|x)\exp\left(r^{weak}(x,y)/\beta\right)$. Then, the optimal policy $\pi^*$ for the above ojective is when $\pi^*(y|x) = \pi^{weak}(y|x) = \pi^{EBM}(y|x)$.*

*Proof.* The optimal solution to the KL divergence in the lower-level objective is minimized when $\pi^{\text{weak}} = \pi^{\text{EBM}}$. substitute this optimal policy in the lower-level objective to the upper-level objective:

$$\min_{\pi_\theta} \text{KL}\left(\pi^{\text{EBM}}||\pi_\theta\right)$$

The solution to this optimization problem is achieved when $\pi_\theta = \pi^{\text{EBM}}$. Which concludes the proof. $\quad\square$

**Theorem 11.2.** *The optimal solution to W2S generalization is equivalent to the optimal solution in the following objective:*

$$\max_{\pi_\theta} \mathbb{E}_{x\sim\mathcal{D}_{weak}, y\sim\pi_\theta(\cdot|x)} \left[r^{weak}(x,y)\right] - \beta KL(\pi_\theta||\pi_{ref}^{weak}) \tag{9}$$

*Proof:* Following Appendix A.1 of Rafailov et al. (2023), we have:

$$\max_{\pi_\theta} \mathbb{E}_{x\sim\mathcal{D}, y\sim\pi_\theta} \left[r(x,y)\right] - \beta\text{KL}\left(\pi_\theta(||\pi_{\text{ref}}^{\text{weak}})\right)$$

$$= \max_{\pi_\theta} \mathbb{E}_{x\sim\mathcal{D}, y\sim\pi_\theta} \left[r(x,y) - \beta\log\frac{\pi_\theta(y|x)}{\pi_{\text{ref}}^{\text{weak}}(y|x)}\right]$$

$$= \min_{\pi_\theta} \mathbb{E}_{x\sim\mathcal{D}, y\sim\pi_\theta(\cdot|x)} \left[\log\frac{\pi_\theta(y|x)}{\pi_{\text{ref}}^{\text{weak}}(y|x)} - \frac{1}{\beta}r(x,y)\right]$$

$$= \min_{\pi_\theta} \mathbb{E}_{x\sim\mathcal{D}, y\sim\pi_\theta(\cdot|x)} \left[\log\frac{\pi_\theta(y|x)}{\frac{1}{Z(x)}\pi_{\text{ref}}^{\text{weak}}(y|x)\exp\left(\frac{1}{\beta}r(x,y)\right)} - \log Z(x)\right] \tag{10}$$

where the normalization factor is:

$$Z(x) = \sum_y \pi_{\text{ref}}^{\text{weak}}(y|x)\exp\left(\frac{1}{\beta}r(x,y)\right)$$

Given an EBM policy defined in Lemma 4.2, we can re-organize objective 10, we have:

$$\min_{\pi_\theta} \mathbb{E}_{x\sim\mathcal{D}, y\sim\pi_\theta(\cdot|x)} \left[\log\frac{\pi_\theta(y|x)}{\pi^{\text{EBM}}(y|x)} - \log Z(x)\right] \tag{11}$$

$$= \min_{\pi_\theta} \mathbb{E}_{x\sim\mathcal{D}} \left[\text{KL}\left(\pi_\theta(\cdot|x)||\pi^{\text{EBM}}(\cdot|x)\right) - \log Z(x)\right] \tag{12}$$

Note that $Z(x)$ does not depend on $\pi_\theta$. Therefore, the RLHF objective is maximized with respect to $\pi_\theta$ if and only if $\pi_\theta(y|x) = \pi^{\text{EBM}}(y|x)$. By Lemma 11.1, we know that the optimal solution to W2S generalization problem is achieved when $\pi_\theta(y|x) = \pi^{\text{EBM}}(y|x)$. Therefore, we have proved that the optimal solution to the RLHF objective and weak-to-stong generalization is equivalent.

## 11.1 Utilizing Strong Student's Knowlege Only for Reward Evaluation

EVE must sample or elicit responses from the strong student during training, to leverage its prior knowledge and balance learning from weak supervision. On the other hand, we could instead use the strong student for reward evaluation of weak supervision data $\mathcal{D}_{\text{weak}}$. Specifically, this results in the following reward function:

$$r^{\text{strong}}(x,y) = \log \pi_{\text{ref}}^{\text{strong}}(y|x) - \log \pi_{\text{ref}}^{\text{weak}}(y|x)$$

and the following optimization objective:

$$\max_{\pi_\theta} \mathbb{E}_{(x,y)\sim\mathcal{D}_{\text{weak}}} \left[ \frac{\exp\left(r^{\text{strong}}(x,y)/\lambda\right)}{Z(x)} \log \pi_\theta(y|x) \right]$$

Intuitively, this objective can be interpreted as a weighted maximum likelihood objective that weighs samples generated by the weak teacher using the implicit reward function of the strong reference model; this allows the training process to scale down undesirable knowledge from the weak reference model $\pi_{\text{ref}}$.

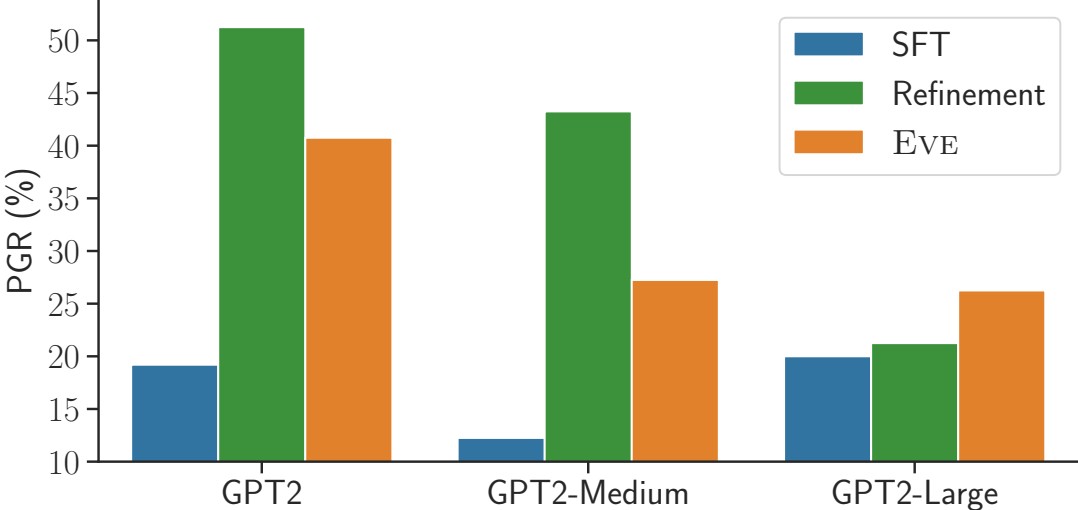

Figure 8: PGR (%) when using the strong student as supervision signal on weak supervision data. We observe that using the strong student as a supervision signal can provide performance improvement compared to SFT.

Fig. 8 illustrates the performance of this approach and the baselines. As can be observed, using the strong student for reward assessment significantly outperforms SFT across different model sizes. This result highlights the substantial benefits of leveraging the strong student in W2S learning. On the other hand, this approach does not outperform Refinement when the weak model is GPT2-Large. Nevertheless, Refinement approaches are more computationally expensive, as they rely on the strong student to "refine" (involves a generation step) data generated by the weak teacher, while this approach utilizes the strong student to only "evaluate" the weak supervision samples.

## 11.2 Comparison with DPO in Controlled Summarization

Tajwar et al. (2024) show that contrastive learning methods, which incorporate negative gradients, outperform offline supervised methods. This is because the negative gradient allows the policy to deviate significantly from its initial distribution when seeking the high reward function, whereas offline supervised methods are unable to guide the policy far from its initial distribution. However, it remains unclear whether this negative gradient provides similar benefits compared to offline supervised approaches in W2S generalization with potentially unreliable supervision.

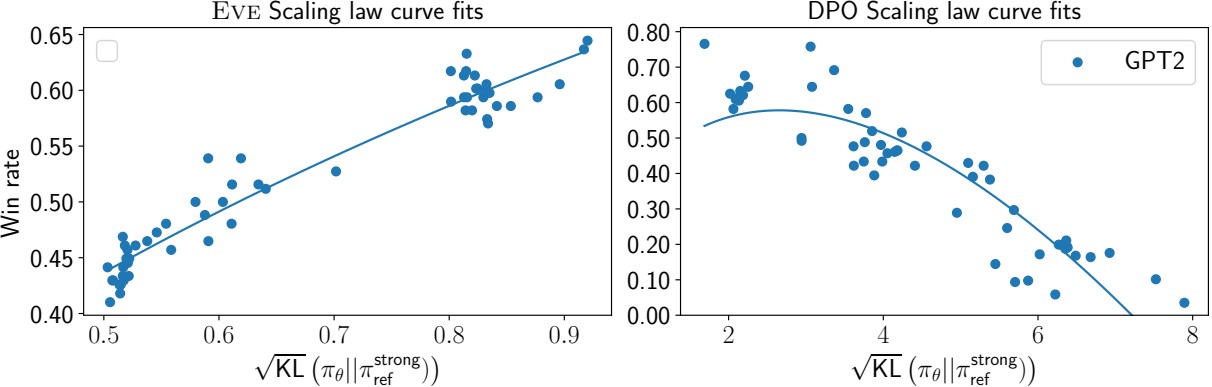

Figure 9: Trade-off between Win-rate and KL-between DPO and EVE methods with varying regularization strength.

Fig. 9 shows the Win rates across different KL-budgets, which corresponds to different values of the KL-regularization hyperparameter, of EVE and DPO. As can be observed, across different weak models, DPO suffers significantly from reward over-optimization, where increasing the KL budget results in a decline in model performance (Rafailov et al., 2024). In the context of W2S generalization, this problem is significant because we cannot easily find an early stopping criterion for training the strong student. In contrast, EVE, an offline supervised approach, does not diverge much from its initial distribution, making it significantly more robust to over-optimization phenomenon across different regularization hyperparameters.

## 12  Sample Refined Responses Across Weak Teachers

| | |
|---|---|
| **Prompt** | SUBREDDIT: r/Advice
TITLE: How to help repay brother?
POST: So my older brother is going to college and he isn't necessarily nice (he's pretty mean and doesn't talk much). He surprised me today by giving me 3 Xbox One games. However, 2 of these games are almost brand new and one is brand new. They are all popular and expensive games and I want to repay him but I don't play Xbox One anymore. I'm still grateful but should I sell them and profit about $150 (I'm only 13 so a lot of money) or should I keep it for sentimental value? I know it sounds bad for me wanting to sell but he's been rude, disrespectful and even abusive once so I don't know what to do. Also should I give it back? Is it the right thing to do? |
| **GPT2** | TL;DR: Brother gives me 3 Xbox One games and I'm really grateful but should I sell them and profit about $150 (I'm only 13 so a lot of money) or should I keep it for sentimental value? I don't play Xbox One anymore. I'm still thankful but should I sell them and profit about $150 (I'm only 13 so a lot of money) or should I keep it for sentimental value? I know it sounds bad for me wanting to sell but he's been rude, disrespectful and even abusive once so I don't know what to do. Also should I give it back? Is it the right thing to do |
| **Strong student** | TL;DR: My brother gave me 3 Xbox One games that are almost brand new and expensive. He is rude and has been abusive. Should I sell it and give him $150 or should I keep it for sentimental value? |

Table 1: Sample prompt, responses from GPT2 teacher response and the strong student after refining.

| | |
|---|---|
| **Prompt** | SUBREDDIT: r/relationships
TITLE: I [M20] met a girl [F20] who i really like while backpacking europe and i dont know what to do.
POST: I met her initially in berlin, where I only got to spend a day with her. I instantly knew there was something between us, I didnt know what but we just clicked strait away. I would have stayed there longer just to be with her if I could have. I met with her again a few months later In london. We spent another 3 days together and I learned we have so much in common. Like a creepy amount in common. Almost all the same interests and hobbies. We had an amazing night out that ended with us sharing a bed and the next day we just spent time together talking. We've arranged to spend another 10 days together next week, just the 2 of us, and I've never been more excited for anything in my life. I havnt been able to stop thinking about her since we last seperated. Were both not the most experienced with relationships (mostly me) so I'm finding it hard to tell her how I feel. But I'm sure I'll find a way next time we meet. I don't even know why Im posting this. Just to help me think I guess. |
| **GPT2-Medium** | TL;DR: met a girl and i thought she was the same person i like. she seemed like the same person i liked. i didn't know what to do. thought i was the same person too but knew it was just a fluke. got to spend 2 days with her in the city and she shared a bed with me. we had amazing night out that ended with us sharing a bed and the next day we just spent time together talking. we've arranged to spend another 10 days together next week, just the 2 of us, and I've never been more excited for anything in my life. I haven't been able to stop thinking |
| **Strong student** | TL;DR: I met a girl who I really like while backpacking europe, we have a lot in common and have arranged to spend more time together next week, I'm not sure how to tell her I like her but I'll find a way next time we meet? |

Table 2: Sample prompt, responses from GPT2-Medium teacher's response and the strong student after refining.

| | |
|---|---|
| **Prompt** | SUBREDDIT: r/relationships
TITLE: I (f/22) have to figure out if I want to still know these girls or not and would hate to sound insulting
POST: Not sure if this belongs here but it's worth a try.
Backstory: When I (f/22) went through my first real breakup 2 years ago because he needed space after a year of dating roand it effected me more than I thought. It was a horrible time in my life due to living with my mother and finally having the chance to cut her out of my life. I can admit because of it was an emotional wreck and this guy was stable and didn't know how to deal with me. We ended by him avoiding for a month or so after going to a festival with my friends. When I think back I wish he just ended. So after he ended it added my depression I suffered but my friends helped me through it and I got rid of everything from him along with cutting contact. Now: Its been almost 3 years now and I've gotten better after counselling and mild anti depressants. My mother has been out of my life since then so there's been alot of progress. Being stronger after learning some lessons there been more insight about that time of my life but when I see him or a picture everything comes back. The emotions and memories bring me back down. His friends (both girls) are on my facebook because we get along well which is hard to find and I know they'll always have his back. But seeing him in a picture or talking to him at a convention having a conversation is tough. Crying confront of my current boyfriend is something I want to avoid. I've been thinking that I have to cut contact with these girls because it's time to move on because it's healthier. It's best to avoid him as well. But will they be insulted? Will they accept it? Is there going to be awkwardness? I'm not sure if it's the right to do and could use some outside opinions. |
| **GPT2-Large** | TL;DR: I'm going through my first real breakup 2 years ago and wound up with a lot of anger and depression from living with my mother and now moving on with my life. Also have a lot of friends who have a lot of respect for my current boyfriend. Will they be insulted by my new boyfriend being in a picture with my exes? Will they accept that? Is there going to be awkwardness? |
| **Strong student** | TL;DR: I (f/22) have to figure out if I want to still know these girls or not and would hate to sound insulting. I've been thinking that I have to cut contact with these girls because it's time to move on because it's healthier. It's best to avoid him as well. But will they be insulted? Will they accept it? Is there going to be awkwardness? I'm not sure if it's the right to do and could use some outside opinions. |

Table 3: Sample prompt, responses from GPT2-Large teacher's response, and the strong student after refining.

