# OpenReview forum: "Weak-to-strong Generalization via Formative Learning from Student Demonstrations & Teacher Evaluation"
_TMLR — Rejected by TMLR_

### Review · Reviewer_wHhc · 2026-01-31

**Summary Of Contributions:**

This paper investigates the Weak-to-Strong (W2S) generalization paradigm where one wants to utilize a weaker (teacher) supervision to elicit the full capabilities of a stronger (student) model. The authors introduce a new method for this setting which they call EVE. This method maximizes the reward while ensuring the policy does not deviate too far from the original model. This is done by including a KL term punishing policy with large differences in distributions from the original strong student.

**Audience:**

No

**Audience Explanation:**

Weak-to-Strong (W2S) generalization is a nascent area of machine learning, with potentially big impact depending how the field moves. I'm sure some of TMLR's audience would find the results and ideas presented in this paper interesting.

**Claims And Evidence:**

No

**Claims Explanation:**

Not yet. There seems to be a few examples of over claiming the significance of the results presented. For example "Fig. 7 also shows the advantages of Eve over contrastive learning approaches (e.g., DPO)". The experiments show better performance over DPO, only.
While the results of the two main experiments performed are very encouraging, and the authors should indeed be proud of their work. I would also be careful with the wording of claims based on these results only. For example I would not agree that paper include extensive empirical results currently. Additionally the plots are constructed in nonstandard way making the different between methods appear bigger. I believe with some minor edits to the text and figures this could be addressed.

**Requested Changes:**

1. Please edit the paper so the claims made reflect the results, there seems to be a few examples of over claiming the significance of the results presented. For example "Fig. 7 also shows the advantages of Eve over contrastive learning approaches (e.g., DPO)". The experiments show better performance over DPO, only. While the results of the 2 experiments performed are very encouraging, I would also be careful with the wording of claims based on these two results only.
"Already capable strong student", should be made more explicit in the abstract introduction as it is a rather important limitation of the work that should be mentioned earlier.
2. Please make plots y axis start at 0.
3. The authors should review the written clarity of the paper as there are a number of typos or words missing, and some sections are a little too brief. Examples:
Random "," under section 2.2,
line before equation 1, KL-constrained... word seems to be missing, also likely reference missing.
sec 7 "the W2S generalization" the "the" here is non-needed.
Sections 3.1, 3.2 and 4.1 should be expanded to make these sections easier to follow for non-experts on this area.
Formative learning - is mentioned a lot but never explained to the reader
4. There is no mention for repeats or any measure of uncertainty in the results, or deviations between runs. Please endeavor to include these for the main results.

---

### Review · Reviewer_f7pM · 2026-02-04

**Summary Of Contributions:**

The paper addresses the "overfitting" problem in Weak-to-Strong generalization. The authors show theoretically that standard fine-tuning acts as implicit regularization toward the weak teacher (Reverse KL), causing the student to mimic teacher's errors. They propose EVE, which flips this: use the weak teacher as a reward model to score the strong student's own demonstrations. This shifts the regularization target back to the student's own prior. Experiments show EVE outperforms SFT and DPO, particuarly in robustness to unreliable feedback.

**Audience:**

Yes

**Audience Explanation:**

Weak-to-strong generalization is relevant to scalable oversight. The paper's framing of this as an implicit regularization problem provides theoretical insight, and the robustness comparison with DPO addresses practical concerns in current RLHF methods.

**Broader Impact Concerns:**

None.

**Claims And Evidence:**

Yes

**Claims Explanation:**

The theoretical link between W2S overfitting and Reverse KL divergence is sound and explains why naive SFT fails. The empirical validation is solid covering appropriate baselines (SFT, Refinement, DPO) across multiple model sizes. The comparison showing DPO's susceptibility to reward over-optimization in this setting (Fig 9) provides strong evidence for the authors' offline supervised formulation.

**Requested Changes:**

The paper has clear theoretical insight and practical method. A few suggestions to strengthen it before the final version:

1. EVE requires sampling K responses from the strong student prior to training to estimate the normalization factor. This is fundamentally more expensive than standard SFT or DPO. It would be helpful to include a brief comparison (e.g., in the Appendix) of wall-clock time or FLOPs for EVE vs. the baselines so practitioners can see the trade-off.

2. In Figure 4, EVE beats the "Refinement" baseline. Since Refinement also relies on the strong student's capability, I want to ensure this is an apples-to-apples comparison regarding the inference compute budget (e.g., number of samples generated). A short clarification would help confirm the gain is due to the objective, not just more sampling.

3. The main results (Figures 3, 4, 5) appear to be single runs. Given the variance in RLHF tasks, it would strengthen the paper to report averages over a few seeds with error bars where possible, especially where margins are tight.

---

### Review · Reviewer_Ko7Q · 2026-02-10

**Summary Of Contributions:**

The paper studies weak-to-strong (W2S) generalization and proposes a method (EVE) that trains a strong student model using samples from a strong reference model weighted by a reward derived from a weak teacher. The authors provide a theoretical framing that interprets weak-teacher imitation as KL-regularized reward maximization toward a weak reference policy, and propose instead to regularize toward the strong reference policy. They derive a weighted supervised objective where weights depend on the likelihood ratio between weak aligned and weak reference models.

Strengths:
- theoretical reinterpretation of weak-to-strong learning through KL-regularized RL
- connects W2S imitation, reward maximization, and forward-KL weighted MLE.
- proposes an offline objective avoiding on-policy sampling from the student.

Weaknesses:
- The core theoretical assumption (weak teacher = KL-regularized RL solution with known reference) is extremely strong and often unjustified.
- The method requires access to a weak reference model \pi^{weak}_ref, which is not available in most realistic W2S settings.
- The derivation relies on fragile importance weighting that appears high-variance and weakly validated.
- Experimental evidence has multiple presentation and fairness gaps. Several baselines are missing or selectively plotted.

**Audience:**

No

**Audience Explanation:**

While W2S generalization is an important topic, the practical applicability of this specific method is limited by its requirements:
- access to both weak aligned and weak reference model likelihoods,
- ability to compute exact log-probabilities,
- assumption that weak supervision arises from KL-regularized RL,
- reliance on importance weighting with potentially unstable ratios.

These conditions do not hold in many real-world alignment or weak supervision settings. As a result, the method is unlikely to transfer broadly. The theoretical framing is also not robust enough to serve as a general foundation for W2S learning, since it depends on a specific generative story for the weak teacher that is rarely verifiable.

This reduces the likely impact for a broad TMLR audience.

**Claims And Evidence:**

No

**Claims Explanation:**

The core theoretical contribution (Theorem 4.2) rests on representing the weak teacher as:
\pi^weak(y|x) = (1/Z(x)) · \pi^weak_ref(y|x) · exp(r^weak(x,y)/\beta).
This assumption is extremely strong and poorly motivated. Why should we believe the weak teacher decomposes this way? Why can we assume we have access to \pi^weak_ref in general? This decomposition only holds if the weak teacher was trained with KL-regularized RL, which is a big assumption.
The theory is a bit limited as can be seen as a tautology: If we assume the weak teacher = KL-regularized RL solution, then imitating it = KL-regularized RL.


The normalization constant Z(x) is intractable and estimated with self-normalized importance sampling from \pi^strong_ref. This introduces potentially high-variance weights and bias. No discussion of when this fails. There is no empirical analysis of:
- weight variance,
- effective sample size,
- sensitivity to K,
- instability due to heavy-tailed ratios.


Experimental evidence is also not fully convincing:
- Figure 3 shows a large performance gap between EVE and SFT before 0.1 epoch, but the starting point at epoch 0 is not shown. This
- DPO and Refinement baselines are missing from that training-curve figure, even though they are central baselines elsewhere. This selective reporting weakens the comparison.
- it is missing statistical significance. There are no error bars, confidence intervals, or significance tests. No multiple runs reported.

**Requested Changes:**

Critical:
- Relax or justify the weak-teacher EBM/KL-RL assumption.
- Provide variance and stability analysis of the Z(x) estimator and weights.
Include effective sample size, weight distribution plots, and sensitivity to K.
- Add fair training-curve comparisons including all baselines (SFT, DPO, Refinement) from epoch 0.
- Add statistical significance testing: multiple runs with different seeds, error bars or confidence intervals, etc.
- Show that lambda/beta = 1 (Fig 5.) is valid for other models also (Qwen or Llama for instance) not only GPT2

Non-critical:
- Compare against simpler reward-weighted regression baselines without likelihood ratios.
- Provide experiments where weak teacher is not derived from RLHF/DPO.
- Add theoretical discussion of bias from self-normalized importance sampling in this setting.

---

### Decision · Action_Editor_vR3H · 2026-03-20

**Recommendation:** Reject

**Additional Comments:**

The reviews are largely consistent in identifying substantial weaknesses in both empirical validation and theoretical grounding, and the lack of author engagement further limits confidence in the work. While the core idea is potentially interesting, the paper would require significant revisions—particularly stronger empirical methodology, clearer justification of assumptions, and improved presentation—to meet TMLR standards.

**Audience:**

Yes

**Audience Explanation:**

Weak-to-strong generalization and scalable oversight are relevant topics that could interest parts of the TMLR audience, and at least one reviewer finds the framing and insights valuable. However, other reviewers argue that strong assumptions and limited practical applicability significantly reduce the broader impact and appeal.

**Claims And Evidence:**

No

**Claims Explanation:**

Multiple reviewers highlight that the empirical evidence lacks statistical rigor (e.g., reliance on single runs without uncertainty estimates) and that key theoretical assumptions are overly strong and insufficiently justified. Additionally, there are concerns about selective reporting and overclaiming relative to the presented results, which together undermine the strength and clarity of the evidence.